# The Impact of Mixed Emotions on Consumer Improvisation Behavior in the Environment of COVID-19: The Moderating Effect of Tightness-Looseness Culture

**DOI:** 10.3390/ijerph192417076

**Published:** 2022-12-19

**Authors:** Xiaozhi Huang, Xiaojie Zhang, Heng Zhang

**Affiliations:** 1School of Business, Guangxi University, Nanning 530004, China; 2School of Management, Lanzhou University, Lanzhou 730000, China

**Keywords:** awe, mixed emotion, knowledge acquisition, tightness-looseness culture, consumer improvised behavior

## Abstract

Organizations and individuals are unprepared for an unexpected outbreak of COVID-19. While most of the literature focuses on improvised reactions at the organizational level, this paper focuses on understanding improvised reactions at the individual level. This paper draws on previous research applying improvisation to the field of consumer behavior and introduces consumer knowledge acquisition as a mediating variable and tightness-looseness culture as a moderating variable from the perspective of mixed emotions of awe and anxiety to explain the mechanism of consumers with mixed emotions of awe and anxiety on improvisation behavior based on the environment of a COVID-19 outbreak. Data from 330 participants in Study 1 examined the effect of mixed emotions of awe and anxiety on improvisation behavior through knowledge acquisition, and data from 434 participants in Study 2 examined the moderating effect of relaxed culture. The findings suggest that consumers with mixed emotions report a higher willingness to acquire knowledge and report higher levels of improvisational behavior. Consumers behaved differently in different environments. Consumers with mixed emotions responded more strongly to improvisation in the loose-culture environment than in the tight-culture environment, and the mixed emotions of awe and anxiety had a positive effect on individual consumers’ improvisational behavior through the mediating role of knowledge acquisition.

## 1. Introduction

When consumers are faced with an unexpected event, they may take some kind of measures to respond [1,2]. If this unexpected event is vastly unknown, consumers may experience emotional changes that in turn affect consumer behavior [3]. Since the outbreak in late 2019, COVID-19 has been spreading rapidly around the world [1], and it is an infectious disease that is highly transmissible through common human activities and is highly contagious and potentially fatal [4]. On 30 January 2020, the World Health Organization declared COVID-19 a “public health emergency of international concern”, and on 11 March 2020, it declared a global pandemic of COVID-19 caused by a novel coronavirus [5]. Although the using of COVID-19 vaccine and related antiviral drugs has provided favorable conditions to combat the epidemic, the journey of the pandemic is not yet over [4], and the global epidemic situation remains risky and uncertain [6].

When low-predictability, high-impact events occur, organizations are faced with situations that change from the routine to the novel, representing a context in which decisions and actions are often made in a context of change, the unexpected and the unforeseen [7]. In the face of this sudden and highly uncertain epidemic, organizations and societies around the world have developed appropriate strategies and interventions [6] that require rapid action by different organizations to contain the spread of the virus as soon as possible and thus prevent further catastrophic consequences of the epidemic [6]. Thus, as a situation shifts from routine to new, flexible field decision-making and informal coordinated responses are required; for example, in the face of the variety of situations that may arise in health systems regarding COVID-19, rigid and stubborn adherence to existing routines may have tragic results [7].

Therefore, improvisation is particularly important in the face of uncertainty, and it has become a way to cope with change and uncertainty [8]. According to Vera and Crossan, improvisational behavior is a new approach to solving objective problems with spontaneity and creativity [9], which is important for any individual [10] and any organization [8]. The related study noted that different organizations have responded differently [7], health systems and organizations need to “proactively adapt and recover from the pandemic” [7], and suggest related measures such as wearing masks and keeping safe distances [4]. The Italian government closed schools across the country for a short time while restricting the movement of people and canceling various non-essential operations [2]. In the country until April 2020, the Chinese government opened some schools and workplaces one after another to restore the previous order [2].

Previous literature points out that the existing literature is understudied as to whether and why the occurrence of a public health emergency such as a COVID-19 outbreak changes consumer behavior [3]. As the environment becomes more complex and unpredictable, each organizational group increasingly values individual improvisation [11,12]. However, most of the current research on improvisation has focused on situational factors and organizational settings [8,12,13,14], with a lack of micro-level individual studies [11]. Related studies have shown that there are individual differences in improvisation development [15], as the process is complex and difficult [10]. When faced with the uncertainty of COVID-19, we know what to do and how to respond at the organizational level, but it is not yet clear how individuals at the micro level develop improvisation skills or generate improvisational behavior [9].

In this paper, based on previous studies that began to explore improvisational behavior in the consumer domain [16] and based on the awe archetype theory and terror management theory, we view the mixed emotions of awe and anxiety as a research perspective and discuss consumer improvisational behavior as an outcome variable in the context of the social environmental conditions in which COVID-19 currently occurs, mediated by knowledge acquisition and moderated by tightness-looseness culture. The relationship between COVID-19-induced mixed emotions and consumer improvisational behavior is discussed with the following theoretical implications and manage-rial insights. (1) Building on previous research, this paper explores the relationship between the mixed emotions of awe and anxiety and consumer improvisation from an affective perspective, which contributes to a deeper understanding of the factors that influence consumer improvisation. (2) By focusing on individual-level improvisation and discussing improvisation in the context of consumer behavior, this paper enriches the literature on consumer improvisation and extends research on the antecedent variables that influence improvisation. (3) Extending some scholars’ views on the strength of social norms and deviance tolerance to micro-consumer behavior research [17], this paper investigates the mechanisms of improvisational behavior of mixed-emotion consumers under different conditions of social norm strength, providing new insights into the mechanisms of improvisational behavior of mixed-emotion consumers. (4) From a management perspective, focusing on the role of mixed emotions in marketing can support firms’ marketing activities and provide theoretical support for specific marketing programs.

## 2. Literature and Hypothesis Development

### 2.1. Mixed Emotions of Awe and Anxiety

Although many scholars have focused on the influence of emotions on consumer behavior for a long time, empirical research on awe was conducted after Keltner and Haidt proposed the awe prototype theory, a theoretical framework that identifies perceived vastness and the need to adapt as two characteristics of awe [18]. The former refers to the fact that the stimulus that triggers awe can be enormous, both physically and conceptually; the latter requires adaptation referring to the fact that awe changes one’s view of the world. Specifically, awe stimulates a need for psychological restructuring to absorb new experiences [18]. In experiencing awe, people’s current psychological frameworks are perceived to be greatly challenged, which in turn promotes adoption or ideological transformation [19]. Awe is an emotion distinct from other positive emotions, elicited by information-rich stimuli, and characterized by cognitive conditioning [20]. It is a typical collective emotion [18] classified as an epistemological emotion characterized by an accompanying shift in consumers’ understanding of the world [21].

Existing research suggests that awe is often described as an experience of self-transcendence in which one can experience beyond everyday concerns and worldly desires [19]. Relevant to this study, awe is also thought to influence consumer behavior by allowing consumers to experience their insignificance and behave in a friendly manner toward society [22], allowing consumers to experience the disappearance of the ego while helping them get along better with the collective [21], making them less likely to value the small things in their daily lives [20,21] and thus reducing their desire for money [19], and enhancing their openness to learning and desire to experience creativity [23].

Considering the consequences and global impact of COVID-19, it is considered a risk event [7]. Related initiatives such as policies to maintain social distance change patterns of social activity [4] and affect consumers’ daily lives. In summary, the crisis of COVID-19 is characterized by both a wide range of perceptions and the need to adapt, so the occurrence of COVID-19 may create a sense of awe [3].

Terror management theory was first proposed by Greenberg to help ensure that individuals were psychologically free from the preoccupation with death [24,25]. Terror management theory considers the coexistence of the awareness of death and the individual’s instinct for self-preservation as a potential cause of anxiety [25], which motivates people to protect themselves from threats by seeking a sense of self-worth and psychological security [26], which in turn enhances the individual’s self-esteem, providing a suitable theoretical framework for understanding potentially life-threatening events [24]. Research has shown that when people are cued to the fragility of life, they experience anxiety [26], and the experiential reminder of death triggers anxiety [25]. It is further hypothesized that this anxiety motivates individuals to make behavioral responses with the expectation of eliminating this insecurity [24].

In contrast to awe, anxiety is defined as a negative emotion that reflects the degree to which an individual consumer associates an uncertain but possible future outcome with being inconsistent with their goals [27,28]. When existing goals are inconsistent with expected future outcomes [27], anxiety can cause consumers to produce certain behavioral responses [29], either from the information received to stimulate seeking and approaching behaviors [27] or avoidance responses [29]. COVID-19 virus is a deadly virus, and in the meantime, as the situation evolves, the mortality rate caused by COVID-19 may vary from day to day and from person to person. When individuals are reminded of death in this deadly crisis, they face fear and experience increased tension, uncertainty, and worry about their current and future lives and livelihoods, all of which can lead to an increase in overall anxiety [26], and the anxiety triggered by COVID-19 induced anxiety may lead individuals to reflect on this crisis, which may result in avoidance and self-protective behaviors [28].

Based on the archetypal concept of awe developed by Keltner and Haidt [18], awe is defined as the emotional experience of wonder that arises when we are confronted with something vast, immense, and beyond our current understanding. Individuals face fear and feel uncertainty and worry about their current and future lives, and all of these emotions can lead to increased anxiety [26]. Although awe and anxiety are two different emotions, they can occur and coexist simultaneously. COVID-19 has a profound impact on our lives and livelihoods, and we cannot know how individuals adapt after a crisis, but assessing the mechanisms that influence the improvised behavior of individuals exposed to a crisis is an important extension of research during an epidemic. This paper argues that consumers will experience both a sense of awe in the face of nature and a sense of anxiety when experiencing COVID-19.

### 2.2. Consumer Improvised Behavior

Improvisation first occurred in theater, music, and other artistic performances and was first introduced into management research by Weick [30], who defined improvisation as the act of responding to an unexpected event without prior arrangement, in which there is only a short interval between planning and execution.

Miner distinguished improvisation from creativity and innovation by arguing that creativity may not include improvisation at all [12], while innovation may be achieved through improvisation but can also be achieved through planning. Cunha argues that there is no good or bad improvisation and that improvisation is a continuous process [8], from low-level interpretation to moderate modification to full improvisation [30]. Scholars have used the concept of improvisation in a variety of situations at the organizational level, including product innovation [12] and new product development [9]. Organizational improvisation starts from the core definition of improvisation, which is “the deliberate integration of designing and executing new work” [12], and it is a process that occurs in teams or throughout the organization [8,12], enabling managers to continuously and creatively adapt to change and continually expand their products and services outward [9].

However, in the current increasingly complex and unpredictable environment, not only do organizations need to respond to unexpected events, but individuals also need to adapt quickly to new environments beyond existing procedures and execution of strategic plans [12]. Therefore, each organizational group is increasingly focusing on the ability of individual improvisation skills [11,12]. Although the act of improvisation is becoming increasingly important, much of the current understanding of improvisation focused on situational factors and the external environment, ignoring the micro-level factors of personal traits, which allow individuals to improvise when the situation demands [9,11]. Vera and Crossan argued that improvisational behavior uses new approaches with spontaneity and creativity to solve objective problems [9]. Thus, improvisation is more of a process [31], and individuals with a high level of improvisation are more likely to jump out of existing constraints, shift their thinking, and make the best use of available resources to solve problems creatively. However, related research has shown that there are individual differences in improvisation development [15] because the process is complex and difficult [10], so how do individuals develop improvisation skills or produce improvisational behaviors [9]? People tend to engage in avoidance behaviors when improvisation involves an element of discomfort and risk [10]. Yet not only do measures need to be implemented quickly, but complex interconnections, which often involve different perspectives, must also be understood [32].

Previous research has shown that improvisation by salespeople affects customer satisfaction [33]; on the one hand, research on salespeople belongs to the management domain, and on the other hand, improvisation measurement and research on salespeople suggest that improvisation can be applied to the consumer field. A different study by Mourey and Felsman further suggests that improvisation training has a positive impact on team writing, self-efficacy, and divergent thinking [34], skills that are critical to modern marketing roles. In the marketing field, the idea that improvisational theatre can be beneficial for the development of marketers is confirmed, illustrating the critical importance of exploring improvisation and its role in marketing events and marketing education [35]. At the same time studies have begun to refer to and examine improvisational behavior in the consumer field. According to the concept of theatrical metaphor [36], consumers are allowed to use the brand environment and its elements as props to improvise and generate their behavior [37]. Focusing on the current COVID-19 and the demands of blockades and social distancing disrupting consumers’ previous shopping habits, consumers are learning to improvise and learn new habits [16]. The authors point out that this is a new area of research and that COVID-19 makes it a research opportunity because of the flexibility of time and the rigidity of place, consumers need to learn to improvise creatively and innovatively to quickly adapt and adopt improvised behaviors in response to government measures [16]. Therefore, this paper draws on Miner’s and Vera and Crossan’s understanding of improvisation to generalize that consumer improvisation refers to consumers’ use of spontaneous and creative new ways to solve objective problems in the face of unpredictable events and to respond immediately to unexpected events [9,12] and discusses improvisation in the field of consumer behavior based on previous research.

### 2.3. Mixed Emotions of Awe and Anxiety and Consumer Improvisation

COVID-19 affects people’s health [4,32], team creativity, performance [38], socioeconomic development [32], and many other aspects, but research on its impact on consumer behavior is still limited [3]. COVID-19 has a profound impact on our lives and livelihoods, and we have no way of knowing how well individuals adapt after a crisis [1,4], but assessing the mechanisms by which individuals are affected by a crisis on improvised behavior is an important extension of the research conducted during the epidemic. Research on emotions suggests that a single emotional experience is not equivalent to awe, nor is it sufficient to produce awe because there may be no breadth of adaptation involved [18]. Thus, unlike other purely singular emotions, awe occurs in conjunction with other emotions in response to the stimulation of a huge object in the physical or conceptual context, while Kim noted that awe emotions are associated with consumer responses [39]. We propose that consumers will experience both feelings of awe and anxiety in the face of nature when experiencing COVID-19 and that one can influence the generation of improvisational behavior by activating the expected feelings of awe and anxiety in marketing communications.

We predicted that consumers would respond to unexpected events in the broader context of the epidemic [1,2], so while better understanding changes over time and environmental factors to explore how individuals relate causality to perceived events or the behavior of others, improvised behavior is even more of an outcome variable that we want to understand. Responding to unpredictable situations requires long-term preparation and additional procedures and capabilities to act in real-time, which is critical for crisis management [2,6], as planning is never sufficient for major events with high uncertainty and ambiguity such as pandemic events [4,7]. In this sense, improvisation is not an arbitrary act but a necessity when time and resources are very limited and future events are unpredictable [40]. Scholars have long emphasized rational consumer behavior and ignored the role of other emotions in influencing individual consumer behavior daily, viewing emotions as one of the factors that limit individual innovation and development [28]. However, at the same time, it has also been pointed out that they are open-minded and think divergently about the problem and are committed to breaking the traditional rules to think and analyze problems from a new perspective, thus adopting different choices than the norm [7,27]. In summary, the following hypothesis is proposed:

**H1:***Mixed emotions of awe and anxiety have a positive effect on consumer improvisational behavior*.

### 2.4. Consumer Knowledge Acquisition

One of the characteristics of awe is adaptation, which drives the need to adapt [18]. Adaptation refers to the fact that when people find discrepancies between what they encounter and their existing knowledge, they must modify or adopt new behaviors to explain and reduce the discrepancy [18,20]. Adaptation is a “stimulus-driven” process [39]. Individuals must change their paradigm to understand the new information contained when they encounter a situation that is different from the previous one. This means that awe creates a sense that one’s current existing knowledge and spiritual structure is inadequate or inaccurate [23], while the experience of wonder induced by awe, for example, is linked to the human instinct of curiosity [20], requiring action to further adapt to the world [18,23]. Further, awe can shake consumers’ confidence in their existing mental structures, making them more open to learning, which in turn stimulates their desire for knowledge acquisition [23]. Some scholars have experimentally demonstrated that the creation of awe makes participants feel that their knowledge is lacking, which in turn stimulates their desire to learn about scientific knowledge [41].

While natural avoidance is the norm when consumers become anxious, current research on anxiety in marketing is usually twofold; anxiety can lead consumers to resort to avoidance behavior [29,42] or take action in the form of approach behavior [27]. The anxiety triggered by the deaths caused by COVID-19 may lead individuals to reflect on this crisis and engage in avoidance and self-protective behaviors [28]. Based on this, we predict that when consumers experience mixed emotions of awe and anxiety, they will be motivated by a desire for knowledge and learning, and we propose the following hypothesis: 

**H2:***Mixed emotions of awe and anxiety have a positive effect on consumer knowledge acquisition*.

In most previous studies, consumers tend to maintain their habitual behavior when experiencing negative emotions such as anxiety or fear [43], while on the contrary, it has been suggested that when faced with uncertainty, consumers tend to seek relevant information from multiple sources before making decisions [44]. Knowledge acquisition is understood as the process by which individuals or organizations acquire external knowledge [45], and consumers will obtain and use information sources in a unique way to reduce uncertainty [44]. It has also been shown that awe stimulates individuals’ desire for knowledge and motivation to learn and will in turn stimulate their desire to respond behaviorally [23], so it is logical to predict that consumers acquire knowledgeable information as a prerequisite for consumers to respond to a given uncertainty [44]. We predict that:

**H3:***Consumer knowledge acquisition has a positive effect on consumer improvisational behavior*.

**H4:***Consumer knowledge acquisition plays a mediating role in mixed emotions of awe and anxiety and consumer improvisation behavior*.

### 2.5. Tightness-Looseness Culture

Most previous cross-cultural research in the field of consumer behavior has focused on cultural values [46], while some scholars have suggested that the normativity of society as a whole may have an impact on consumer behavior [47]. In this paper, we draw on Gelfand in extending consumer behavior research through the strength of social norms and tolerance for deviance [17]. Tightness-looseness culture theory originally originated from anthropological studies of traditional societies [48], and although few studies have explored the impact of tightness-looseness culture on consumer behavior in the marketplace, a growing number of scholars have called attention to the untapped potential of this theory [17,46,47,49]. Looseness and tightness refer to differences in the strength of norms and tolerance of norm deviations across human groups [50], with the strength of norms referring to the unwritten rules and social pressures that individuals feel compelled to follow in a given culture and tolerance referring to the severity of a punishment that results when individuals violate norms [51].

According to tightness-looseness culture theory, looseness culture environments have less strongly enforced social norms and a higher tolerance for deviant behavior than tightness culture environments [50]. Research on the interaction of norms and values can also provide new insights into how consumers respond to norm violations in the marketing arena [47]. Consumers in tightness cultures are more disturbed when they violate norms than those in looseness cultures, but their reactions may vary depending on cultural values [17]. 

Cultural tightness and looseness can be borrowed from the psychological measures of one’s attitudes toward social norms [50]; however, given that cultural closeness is a social adaptation to threats, some scholars expect it to have a significant impact when potential threats are prominent [52]. It has been suggested that safety during pandemics can be considered in the context of the cultural closeness of consumer behavior [17,47] and that the awe triggered by COVID-19 may also motivate people to comply with social norms [3]. Then, after experiencing COVID-19, we focus on improvised behaviors triggered by changes in consumer emotions by linking emotions to tightness-looseness culture and discussing the different conditional effects of mixed emotions of awe and anxiety in different environments of cultural normative intensity. We predicted that the mixed emotions of awe and anxiety would differentially influence consumer improvisation behavior in both looseness culture and tightness culture environments, specifically proposing the following hypothesis:

**H5:***Tightness-looseness culture moderates the relationship between mixed emotions of awe and anxiety and consumer improvisational behavior. In conditions of higher social tolerance (looseness culture environment), consumers holding mixed emotions report higher levels of improvisational behavior. Conversely, in conditions of low social tolerance (tightness culture environment), consumers have less impact of mixed emotions on consumer improvisation behavior*.

### 2.6. Theoretical Model

Firstly, existing research confirms that terror management theory proposes that when a potentially life-threatening event occurs, the reminder of the death experience of that event triggers anxiety in consumers [25], and terror management theory provides an appropriate theoretical framework for understanding potential life-threatening events [24]. Against the backdrop of COVID-19 outbreak that influenced the world in 2020, and based on the transmissibility and high lethality of COVID-19 virus, this paper considers the mixed emotions of awe and anxiety as the starting point for a study of consumer behavior at the micro level in the face of unpredictable events.

Secondly, according to the awe prototype theory [18], when individuals encounter information that does not meet current expectations, they must change their existing ways and absorb this new information. In this context, it has been proposed that such a unique experience leads to a greater awareness of one’s knowledge gap [41], and a possible outcome of recognizing this gap is the desire to take relevant action to fill the gap that exists [23,41], thus predicting that consumers experiencing a mixture of awe and anxiety may positively influence consumer improvisational behavior (H1) through the mediating role of knowledge acquisition (H4).

Further, most of the original studies discussed improvisational behavior in the domain of organizational management [10,53], this paper discusses improvisational behavior in the consumer domain based on the original studies [16,37]. Koivisto and Mattila stated that according to the concept of theatrical metaphor [37], consumers can improvise using the brand environment and related elements as props and that consumers whose consumption habits are disrupted are developing new habits and learning to improvise [16]. In this paper, we shift our perspective to the field of consumer behavior to discuss the incorporation of improvisation and discuss the relationship between mixed emotions and consumer improvisational behavior.

Subsequently, it is known from existing studies that COVID-19 is associated with tightness-looseness culture [38]. Relevant scholars have defined tightness-looseness culture as the strength of social norms and tolerance for norm deviation [50], and when COVID-19 poses a significant threat, members of the same social organization need to share social norms to ensure order and take preventive measures [38]. Past research on tightness-looseness culture has focused almost exclusively on the national level [38,50], and we draw on some scholars’ inclusion of tightness-looseness culture in influencing consumer behavior [17,46,47] to discuss the influence of mixed emotions on consumer improvisation behavior in tightness-looseness cultural environment mechanism.

In summary, we have attempted to provide insights into these issues by drawing on terror management theory and tightness-looseness theory to develop a framework model throughout the text to understand the impact of the mixed emotions of awe and anxiety experienced by individuals in the current epidemic on consumers’ improvisational behavior. We predict that mixed emotions of awe and anxiety have a positive effect on consumer improvisation behavior, and the mediating effect through knowledge acquisition remains. At the same time, the mixed emotions of awe and anxiety under the moderating effect of tightness-looseness culture have different effects on consumers’ improvisational behavior. In summary, the model framework of this paper is shown in Figure 1.

## 3. Study 1

### 3.1. Study Content

Study 1 aimed to examine the main effects and mediating mechanisms of mixed emotions of awe and anxiety on consumer improvisation behavior (H1, H2, H3, and H4), whether mixed emotions of awe and anxiety have a direct effect on consumer improvisation behavior, and whether consumer knowledge acquisition plays a mediating role in mixed emotions and consumer improvisation behavior. For the responses to the above questions, Study 1 used a questionnaire to collect data on each variable and analyzed the data using SPSS 26.0 (IBM Corp., Armonk, NY, USA).

Data for Study 1 on mixed emotions, consumer knowledge acquisition, and con-sumer improvisational behavior were collected from 14 September 2021, and a total of 99 valid data were received over three days. A secondary data collection exercise was conducted during a three-day data collection period from 6 August 2022, resulting in a total of 355 questionnaires for the final Study 1. 

All data was collected on China’s largest online survey platform, Wen Juan Xing (Changsha Ranxing Information Technology Co., Ltd., Changsha, China). Wen Juan Xing (https://www.wjx.cn/, accessed on 6 August 2022), which is equivalent to Qualtrics, SurveyMonkey, or CloudResearch, provides online questionnaire design and survey capabilities for companies, research institutions, and individuals. Questionnaire Star’s sample database covers more than 2.6 million respondents, whose personal information is validated to obtain a real, diverse, and representative sample.

A total of 355 participants took part in the experiment, and 330 valid questionnaires were returned with an effective rate of 92.96%, including responses from a total of 169 females and 161 males, M_age_ = 28.54, SD = 7.62. Participants were randomly assigned to the mixed emotion group (165) and the neutral group (165) to complete the questionnaire in question. The questionnaire for each variable was based on a well-established scale developed and widely used by domestic and foreign scholars, and the actual situation was rated on a 7-point Likert scale, with 1 indicating “not at all” to 7 indicating “completely”.

### 3.2. Study Design

Participants were first manipulated for emotion regarding Galoni et al.’s picture and text stimulation method [43]. Participants in the mixed emotion group of awe and anxiety viewed a set of photographs and read information about COVID-19 virus during COVID-19 outbreak before completing the questionnaire, and participants in the neutral group were not manipulated. The stimulation of awe was based on Galoni et al.’s picture and text stimulation method [43], with a set of photos related to COVID-19, such as ordinary heroes helping each other, angels in white going out on orders, and relatives dying and still standing on the front line. The material on anxiety stimulation refers to the textual description method used by Huang and Sengupta [54] and Galoni et al. [43], which describes the transmissibility and very high lethality and confirms the existence of human-to-human transmission of the virus, with respiratory and close contact being the main routes of transmission. After the completion of the emotional stimulus manipulation, the participants filled out a questionnaire to report their current emotional state. The Awe Scale was adopted from the Trait Awe Scale developed by Piff et al. [22]. The Anxiety Scale uses the Self-Rated Anxiety Scale developed by Zung [55]. This was followed by a measurement of consumer improvisation, which was based on the Individual Improvisation Questionnaire developed by Vera and Crossan [9]. The specific measurement scales are shown in the following Table 1.

### 3.3. Hypothesis Testing and Discussion of Results

#### 3.3.1. Manipulation Test

Referring to Galoni et al.’s picture and text stimulation method for emotional manipulation of participants [43]. Participants in the mixed emotion group of awe and anxiety viewed a set of photographs and read information about COVID-19 virus during COVID-19 outbreak, for the neutral group no manipulation was performed. A total of 330 participants participated in this study, and participants were randomly assigned to the mixed emotion group (165 participants) and the neutral group (165 participants) to complete the questionnaire, while the rest of the participants’ data were removed for failing the test items. After the emotion manipulation, the participants filled in the related awe and anxiety emotion scale. Next, they were asked to fill in the mediating variable knowledge acquisition and the dependent variable consumer improvisation behavior scale according to their feelings, which were both 7-point Likert scales, with 1 representing “not at all in line with their feelings” and 7 representing “completely in line with their feelings”. One-way ANOVA results showed that participants in the mixed emotion group had higher feelings of awe (M = 5.85, SD = 0.93) and anxiety (M = 5.27, SD = 1.17) than those in the neutral group (M = 5.25, SD = 1.04) and anxiety (M = 3.37, SD = 1.47), the results are shown in Figure 2. The independent samples *t*-test showed t = 14.7, 95% CI = (11.89, 15.56), and there was a significant difference in emotion between the mixed emotion group and the neutral group, so the manipulation test for emotion was successful.

#### 3.3.2. Main and Mediating Effects Test

For the dependent variable consumer improvisational behavior, the main reference is the Individual Improvisational Behavior Questionnaire developed by Vera and Crossan [9], which includes seven questions with specific items, such as “able to deal with unanticipated events immediately”, measuring Cronbach’s alpha coefficient of 0.88 and KMO and Bartlett’s test coefficient of 0.90. An independent samples *t*-test, which is a method of testing whether there is a significant difference between the means of two mutually independent groups of samples, was used to test the effect of emotions on consumer improvisation behavior. The results showed that mixed emotions had a facilitative effect on consumer improvisation behavior, which was higher in the mixed emotions group (M = 5.60, SD = 0.81) than in the neutral group (M = 5.07, SD = 1.01), F (1, 328) = 1.83, *p* < 0.05, t = 5.25, 95% CI = (0.33, 0.73), *p* < 0.05, verifying the hypothesis that the mixed emotions of awe and anxiety positively affects consumer improvisation behavior and that the mixed emotions of awe and anxiety better promotes consumer improvisation behavior.

The mediating role of consumer knowledge acquisition under the effect of mixed emotions of awe and anxiety on consumers’ improvisational behavior was then examined. Cronbach’s alpha coefficient for the consumer knowledge acquisition measure was 0.86, and the KMO, and Bartlett’s test coefficient was 0.74. The Process plug-in for SPSS is an add-in for the analysis of mediating and moderating effects, providing dozens of models. Simply select the corresponding model and set the corresponding independent, dependent, mediating, or moderating variables to produce the desired results. This study was analyzed using Model 4 in the PROCESS plug-in in SPSS software [56], tested using the Bootstrap method, with a sample size of 5000 selected, a bias-corrected nonparametric percentile method selected for the Bootstrapping sampling method, and Model 4 selected for the mediation model with a 95% confidence interval.

The results are shown in Table 2 and Figure 3. The path of the effect of mixed emo-tions of awe and anxiety on consumer knowledge acquisition was β = 0.0272, SE = 0.0046, t = 5.8692, *p* < 0.01, 95% CI [0.0181,0.0363], which verified hypothesis 2 that mixed emo-tions of awe and anxiety have a positive effect on consumer knowledge acquisition. β = 0.4976, SE = 0.0494, t = 10.0827, *p* < 0.05, 95% CI (0.4005,0.5947) for the consumer knowledge acquisition-consumer improvisation behavior influence path, verifying hypothesis 3 that consumer knowledge acquisition has a positive influence on consumer improvisation behavior. The β = 0.0026, SE = 0.0044, t = 0.5975, *p* > 0.05, 95% CI (–0.0060,0.0112) of the mixed emotion-consumer improvised behavior influence path of awe and anxiety contains 0, indicating that the direct effect is not significant. However, the overall effect value of the independent variable on the dependent variable was 0.0161, SE = 0.0047, t = 3.4077, *p* < 0.05, 95% CI (0.0068,0.0254), which, combined with the results of the direct effect is known to be a full mediating effect, and Hypothesis 4 states that consumer knowledge acquisition mediates between the mixed emotions of awe and anxiety and consumer improvisational behavior. 

Specifically, the mixed emotions of awe and anxiety promote the generation of consumer improvisational behavior, and this influence mechanism will be mediated through consumer knowledge acquisition, i.e., the mixed emotions of awe and anxiety promote consumers’ willingness to acquire knowledge, and consumer knowledge acquisition positively influences the generation from consumer improvisational behavior.

#### 3.3.3. Results Discussion

Study 1 discussed the influence of mixed emotions on consumer improvisation behavior and the mediating role of consumer knowledge acquisition. In Study 1, this paper considers the awe emotions [3] that would be elicited in COVID-19 with the anxiety emotions [25] that may be elicited based on terror management theory into a theoretical model that discusses the impact of mixed emotions on consumers’ improvisational behavior mediated by knowledge acquisition. Most of the existing research on improvisation has focused on the field of organizational management [10], and this paper draws on previous academic research to apply improvisation to the field of consumer behavior [16,37]. The results found that consumers in the mixed emotion group reported higher levels of consumer improvisation compared to the neutral group, while consumer knowledge acquisition did mediate between mixed emotion and consumer improvisation. Previous literature indicates that awe enhances consumers’ openness to learning and desire to experience creativity [16,37] and willingness to change existing patterns to learn from the outside to receive new knowledge [39]. Study 2 further verifies the relationship between mixed emotions and consumer improvisation and the mediating role of knowledge acquisition, and on this basis, discusses the influence of tightness-looseness culture on the relationship between both mixed emotions and consumer improvisation.

## 4. Study 2

### 4.1. Study Content

Existing research has demonstrated the effectiveness of tightness-looseness culture in the Chinese context [57]. The purpose of Study 2 was to examine the moderating effect of tightness-looseness culture on the effect of mixed emotions of awe and anxiety on consumers’ improvisational behavior, i.e., consumers in looseness culture felt higher mixed emotions than in tightness culture and thus promoted higher consumer improvisational behavior. This is because individual consumers are likely to produce more improvisational behavior in a relaxed cultural environment, where intergroup norms are less intense and tolerance for norm deviation is greater.

Data for Study 2 on the mediating role of mixed emotions on consumer improvisation behavior through consumer knowledge acquisition in different cultural environments (looseness-culture and tight-ness-culture environments) were collected about three days from 15 September 2021, with 191 valid questionnaires received. A secondary data collection exercise was conducted on 6 August 2022, resulting in a total of 451 valid questionnaires for Study 2.

A total of 451 participants participated in the experiment, the sample participants were all Chinese citizens. A total of 17 people failed the attention test questions, so their questionnaires were excluded from the analysis. A total of 434 valid questionnaires were recovered, including responses from 273 females and 161 males, and the questionnaire recovery rate was 96.23%. 

### 4.2. Study Design

Study 2 conducted 2 (emotion: mixed emotion group vs. neutral group) × 2 (tightness-looseness cultural environment: looseness cultural environment vs. tightness cultural environment) was used for analysis, and participants were randomly assigned to complete the questionnaire as follows: the mixed emotion-looseness cultural environment group (114 participants), mixed emotion-tightness cultural environment group (105 participants), neutral-looseness cultural environment group (108 participants), and neutral-tightness cultural environment group (107 participants) were completed, M_age_ = 27.08, SD = 7.04.

Under the premise of also manipulating emotions for different cultural situations, tightness-looseness culture was manipulated using the textual description method proposed by Gelfand et al. [50] and using their proposed scale (see Table 1) into different cultural groups of participants reading different contextual information. Participants filled out the Looseness-Tightness Culture Scale and the Improvisational Behavior Scale after reading the experimental contextual material to measure the level of consumer improvisational behavior.

### 4.3. Hypothesis Testing and Discussion of Results

#### 4.3.1. Manipulation Test

The variable on tightness-looseness culture was manipulated using the textual description method proposed by Gelfand et al. [50] where participants were randomly entered into groups and then read information about tightness-looseness culture, respectively.

Participants who entered the looseness culture group read the following information—Many historians believe that Country Y was so successful because it was built on a foundation of freedom and openness. Country Y effectively prevents oppression through freedom of expression for all, had few and weak social norms, low penalties for norm violations, and prevented persecution through diversity and inclusive values.

Participants who entered the tightness culture group read the following information—Many historians believe that Country Y was so successful because it was built on a foundation of law and order. Country Y had a large organized and regulated police system that effectively prevented crime and the spread of disease through effective travel and quarantine and had many strong social norms and strong penalties for violations. Cronbach’s alpha coefficient measured by the Looseness-Tightness Culture Scale was 0.82, and the KMO and Bartlett’s test coefficient was 0.89. Participants filled in the Improvised Behavior Scale after reading the experimental contextual material to measure consumer improvised behavior.

For the manipulation of emotions as in Study 1, the results of the one-way ANOVA showed a significant difference in emotion between the mixed emotion (M = 5.43, SD = 0.87) and the neutral group (M = 4.23, SD = 1.17), F (1432) = 130.58, *p* < 0.05. For the results of the tightness-looseness variable, there was a significant difference between the looseness culture group (M = 4.59, SD = 1.14) and the tightness culture group (M = 5.49, SD = 0.68), F (3430) = 36.67, *p* < 0.05, indicating that the emotion and tightness-looseness culture were manipulated successfully in both groups.

#### 4.3.2. Mediated Tests with Moderation

To more intuitively demonstrate the mediating effect of knowledge acquisition under different cultures, this paper draws on the mediation effect analysis method of within-subject design proposed by Montoya and Hayes [58], refers to the sequential analysis method and path analysis procedure suggested by Wang and Wen [59] and others, and uses a MEMORE plug-in in the SPSS software to analyze the mediating effect of knowledge acquisition, using 5000 replicate samples. The MEMORE plug-in extends the extrapolation of indirect effects widely used in inter-participant designs to this intra-participant mediation analysis. Using this path analysis approach, comparisons of indirect effects in more complex models with multiple mediators running in parallel and serial can be discussed. Therefore, the MEMORE plug-in can be used to perform two-condition within-participant statistical mediation analysis. This paper uses the MEMORE plug-in to further examine the mediating effects of consumer knowledge acquisition in a looseness-culture environment and a tightness-culture environment.

The model-related variables were set as follows: the independent variable was the mixed emotion of awe and anxiety, the mediating variable was knowledge acquisition, and the dependent variable was consumer improvisational behavior. The results of the mediation test for knowledge acquisition are shown in Figure 4 and Figure 5.

Figure 4 gives the results of the analysis of the mediating effect of knowledge acquisition in the looseness culture environment. From Figure 4, it can be seen that for the looseness culture environment, (1) The results of the paired samples t-test showed significant differences in the means of knowledge acquisition between the two environments in the mixed emotion and neutral groups, i.e., a = ΔM = M_mixed emotions group_ − M_neutral group_ = 0.2815, t = 3.145, *p* < 0.01; and significant differences in the means of consumer improvisation, ΔM = M_mixed emotions group_ − M_neutral group_ = 0.3423, t = 3.062, *p* < 0.01. The mixed emotions group had higher levels of knowledge acquisition than the neutral group, M_mixed emotions group_ = 6.2685 and M_neutral group_ = 5.9870; consumer improvisational behavior was also higher, M_mixed emotions group_ = 5.588 and M_neutral group_ = 5.2456. (2) The path coefficient of the difference between the mean value of knowledge acquisition to the mean value of consumer improvisation behavior in the mixed emotion and neutral groups was significant, b = 0.4578, t = 4.0651, *p* < 0.001; the effect of mixed emotions on consumer improvisation behavior was significant, c′ = 0.2135, t = 1.9718, *p* > 0.05. (3) The non-parametric percentile Bootstrap method was used to directly test the mediating effect of knowledge acquisition (the significance of a × b), and the results showed that the 95% confidence interval of a × b was [0.0390, 0.2412] without 0, indicating that knowledge acquisition plays a significant mediating role in the relationship between mixed emotion and consumer improvisation in a tightness-looseness environment, and H4 was validated.

Figure 5 gives the results of the analysis of the mediating effect of knowledge acquisition in the tightness culture environment. From Figure 5, it can be seen that for the tightness culture environment, (1) The results of the paired samples *t*-test showed that the means of knowledge acquisition in the two scenarios differed significantly between the mixed emotion and neutral groups, i.e., a = ΔM = M_mixed emotions group_ − M_neutral group_ = 0.2349, t = 2.130, *p* < 0.05; the means of consumer improvisation differed significantly, ΔM = M_mixed emotions group_ − M_neutral group_ = 0.0.3374, t = 2.981, *p* < 0.01. Mixed emotions had higher levels of knowledge acquisition than the neutral group, M_mixed emotions group_ = 5.8634 and M_neutral group_ = 5.6285; consumers also had higher levels of improvisational behavior, M_mixed emotions group_ = 5.2898 and M_neutral group_ = 4.9523. (2) The path coefficient of the difference between the mean value of knowledge acquisition to the mean value of consumer improvisation behavior in the mixed emotion and neutral groups was significant, b = 0.5546, t = 6.2655, *p* < 0.001; the effect of mixed emotion on consumer improvisation behavior was significant, c′ = 0.2071, t = 2.0853, *p* < 0.05. (3) The non-parametric percentile Bootstrap method was used to directly test the mediating effect of knowledge acquisition (the significance of a × b), and the results showed that the 95% confidence interval of a × b was (0.0100, 0.2662), which did not contain 0, indicating that knowledge acquisition plays a significant mediating role in the relationship between mixed emotion and consumer improvisation in the tightness culture environment, and H4 was validated.

#### 4.3.3. Moderating Mechanism Test

The tightness-looseness culture scale was used by Gelfand et al. [50] to measure the degree of tightness-looseness culture perceived by consumers using six scales including that people in country Y should follow many social norms and that people in country Y almost always follow social norms.

Using emotion (mixed emotions vs. neutral) as a grouping variable, a one-way ANOVA with an independent samples *t*-test was conducted on consumer knowledge acquisition and consumer improvisation behavior in a looseness culture environment and a tightness culture environment. The results showed that the knowledge acquisition of consumers with mixed emotions in a looseness culture environment (M = 6.28, SD = 0.76) was higher than that of the neutral group (M = 5.99, SD = 0.76), F (1220) =9.48, *p* < 0.05, again verifying that mixed emotions have a facilitating effect on consumer knowledge acquisition. Also, consumer improvisation behavior in the mixed emotion group (M = 5.60, SD = 0.97) was higher than consumer improvisation behavior in the neutral group (M = 5.24, SD = 0.88), F (1220) = 8.02, *p* < 0.05, validating this paper’s prediction that mixed emotions promote higher consumer improvisation behavior in a looseness culture environment. The improvisation behavior of consumers in the mixed emotion group in the tightness culture environment (M = 5.29, SD = 0.91) was lower than that of mixed emotion consumers in the looseness culture environment (M = 5.60, SD = 0.97) compared to the looseness culture environment, F (1217) = 5.87, *p* = 0.016 < 0.05, supporting the hypothesis.

The PROCESS plug-in model1 was used to test the moderating role of tightness-looseness culture in the relationship between mixed emotions of awe and anxiety and consumer improvisational behavior and to analyze the interactive effects of mixed emotions and tightness-looseness culture on consumer improvisational behavior, with a Bootstrapping sample size of 5000 and a 95% confidence interval. The results are shown in Figure 6 showing the significant interaction effect of mixed emotions and tightness-looseness culture on consumer improvisation behavior, SE = 0.0951, t = −1.9854, *p* = 0.0477 < 0.05, 95% CI (−0.3756, −0.0019).

The mediating and moderating effects of the overall framework proposed in this paper were then tested again using the PROCESS plug-in model5 and the results are shown in Table 3 below. Hypothesis 1’s main effect of mixed emotions of awe and anxiety on consumer improvisation behavior and Hypothesis 5’s moderating effect of tightness-looseness culture between mixed emotions of awe and anxiety and consumer improvisation behavior was verified, i.e., the looseness cultural environment was more likely to promote the positive effect of mixed emotions of awe and anxiety on consumer improvisation behavior more than the tightness cultural environment.

#### 4.3.4. Results Discussion

Study 2 added tightness-looseness culture as a moderating variable for discussion based on Study 1, specifically discussing the impact of mixed emotions in different contexts on consumer improvisation behavior, responding to the previous scholar on the inclusion of tightness-looseness culture for consideration at the individual level [17] and discussing the impact of the interaction of mixed emotions and cultural context on consumer improvisation behavior. This paper builds on previous studies that have included improvisation in the field of consumer behavior [16] by considering mixed emotions and knowledge acquisition in conjunction with tightness-looseness culture to further substantiate the mechanisms of influence on consumer behavior. The results of Study 2 reconfirm that mixed emotions promote consumers’ willingness to acquire knowledge, and after including tightness-looseness cultures for consideration, this paper finds that mixed-emotion consumers in looseness culture environments report higher willingness to acquire knowledge and higher levels of consumer improvisation than mixed emotion consumers in tightness culture environments.

## 5. Conclusions and Discussion

### 5.1. Research Findings

Based on the awe archetype theory and terror management theory, this paper focuses on linking the mixed emotions of awe and anxiety experienced by consumers in the current social environment of COVID-19 to individual consumer improvisation behaviors and discusses the possible mechanisms of mixed emotions’ influence on consumer improvisation behaviors instead of the previous studies, for example, on consumer adoption of new products [27], typical vs. atypical product choices [54], or preference behavior for familiar goods [43].

The study conducted questionnaires and laboratory experiments with 764 participants to test the research model and related hypotheses of this paper. Specifically, this paper demonstrates through two studies that the combined effect of mixed emotions of awe and anxiety enhances consumers’ desire to acquire knowledge and promotes improvisational behavior and that the mediating effect of consumer knowledge acquisition holds. There are differential effects of mixed emotions of awe and anxiety on consumer improvisation behavior under the moderating effect of tightness-looseness culture, specifically, in a looseness culture environment, consumers with mixed emotions report greater consumer improvisation behavior, findings that contribute to theory and management practice.

### 5.2. Theoretical Contributions

The research in this paper has the following main theoretical contributions.

The paper builds on previous empirical studies on the effect of awe on consumer behavior [19,22], in the context of the current environment of COVID-19, and draws on the view that death reminders stimulate anxiety in consumers as proposed by terror management theory [25,26] to investigate the mixed awe and anxiety emotions to make relevant research. While considering the impact of mixed emotions on consumers in their daily lives [27,54], our findings contribute to enriching the literature on the psychological and behavioral impact of COVID-19, contributing to the study of mixed emotions and enriching the emotional consumption literature in this regard. We demonstrate that mixed emotions of awe and anxiety can promote consumer improvisational behavior, and the work in this paper provides a new perspective on motivating consumer improvisational behavior and offers a new mechanism to explain the relationship between mixed emotions and improvisational behavior.The research in this paper contributes to an enhanced understanding of individual knowledge acquisition, as awe is a new method of enhancing people’s access to knowledge, a finding that has theoretical contributions to marketing. Studies have noted that consumers often avoid learning, change and other forms of mental activity [19,21] because individuals hold certain inertia prone to maintaining established habits [23]. To overcome these barriers, marketers often use behavioral influence strategies (e.g., promotions or waiting for consumers to experience life role transitions) [27,29], but such strategies do not address the underlying problem: consumers lack intrinsic motivation to learn [23]. The findings of this paper contribute to a specific understanding of the domain of mixed emotions and knowledge acquisition, and we have identified a factor that can be further explored that can influence consumers’ desire for knowledge acquisition. This study adds to previous work that found that awe motivates consumers’ knowledge acquisition and thus leads to improvisational behavior, revealing a key prerequisite for improvisational behavior.Previous work has provided valuable insights into improvisation at the organizational level [8,10]; our study adopts a different perspective, shifting improvisation research from the organizational to the individual level, considering improvisation at the field of consumer behavior, enriching the literature on improvisation at the individual level, and expanding the antecedents of improvisation dependent variable research. Also, taking tightness-looseness cultural variables into account extends previous research on mixed emotions by focusing on how they can motivate the generation of consumer improvisation and whether tightness-looseness cultural affects them differently and provides evidence of a new factor that can enhance consumer improvisation. It also provides empirical evidence on how consumers improvise in the face of unexpected events under the condition of tightness-looseness, which in turn provides support for remedial measures by business organizations. Integrating improvisational behavior into the emotional event theory framework and the tightness-looseness culture theory framework for research enriches the literature on the influence of emotional and cultural environments on consumer improvisational behavior and also extends the research on the antecedent variables affecting improvisational behavior and the outcome variables of tightness-looseness culture.By linking improvisational behavior in the face of unexpected events to tightness-looseness culture, our study focuses on consumer behavioral responses influenced by external tightness-looseness culture factors in the face of unpredictable events at the individual level, and further discusses the mechanisms by which tightness-looseness culture affects consumer improvisational behavior. We respond to the call of Gelfand to consider the tightness-looseness culture at lower levels such as team and individual levels [17,50]. The findings in this paper, therefore, help to reveal that tightness-looseness culture can also be differentially influenced at the individual level. In addition, past research has shown that tightness-looseness culture is a double-edged sword for some macro social indicators at the national level [48], our work also echoes these findings from a micro perspective, where the mixed emotions of awe and anxiety elicited by COVID-19 have both positive (promoting improvisation) and negative (inhibiting consumer improvisation) aspects under the influence of tightness-looseness cultural factors, and these findings suggest future research directions to further discuss the impact of tightness-looseness culture formation at the micro level.

### 5.3. Management Implications

Awe as an emotional theme that continues to be valued [19,22,23] has been shown that marketers can conduct sales activities by eliciting awe [60], and valuing its valuable role in the marketing field can provide support to the business and marketer side, providing theoretical support for practical marketing programs [23,61]. The research on the motivational effects of anxiety in marketing is mixed; on the one hand, anxiety may cause consumers to develop avoidance behavior, making them want to move away from the stimulus that induces anxiety [29]; on the other hand, anxiety may also cause consumers to develop an approach behavior that leads to certain actions [27]. From a management perspective, if market research shows that consumers have a strong sense of anxiety, then marketers can start with communications that evoke a sense of awe, rather than trying to downplay the anxiety.Our findings suggest that when faced with unexpected situations, a mixture of awe and anxiety drives consumers to improvise. So, in marketing, if the market is launching a new product or promoting an existing product, it is necessary for marketers to communicate in a way that evokes awe and anxiety, rather than trying to downplay it. Although in this paper, we do not test whether advertisements that evoke mixed emotions of awe and anxiety affect consumer purchase behavior, previous research has established that mixed emotions of hope and anxiety, and mixed emotions of disgusted and fear affect consumer behavior. Further, when consumers feel anxious about their surroundings, advertisements can be placed with photos or videos of magnificent natural scenery that evoke feelings of awe [19,22], and consumers may be more likely to generate consumer behavior after eliciting mixed emotions than those who do not have mixed emotions.Previous research findings suggest that improvisation by marketers can increase consumer satisfaction [38]. Then, in the face of different groups of consumers, marketers can generate improvised behavior and thus improve consumer satisfaction, and management can provide appropriate training for marketers to develop their independent thinking and action skills in response to unexpected events. In a competitive and unpredictable market, marketing managers should encourage marketers to improvise more. On this basis, marketers can use the experience of the outside environment as an entry point for consumers and can remove non-essential regulations upon entering the store to allow consumers to experience a higher tolerance for deviations from the norm in the outside environment and thus promote improvisational behavior.

### 5.4. Research Limitations and Future Research Directions

This paper uses the experimental method to examine the effect of mixed emotions of awe and anxiety on consumers’ improvisational behavior. In terms of emotions, the specific assessment dimensions that consumers experience and perceive their own emotional experiences in daily life and the questionnaire that distinguishes discrete emotions may be different. In terms of emotion measurement, the scale is filled in after stimulus manipulation using word pictures, simple manipulation and a single scale are not the only way to measure emotions. While it is useful to understand the differences in emotions when studying discrete emotions in isolation, discrete evaluation dimensions may not be diagnostic of emotional experience, and understanding mixed emotional states is critical to understanding how the overall assessment of emotions influences behavioral tendencies and consumption behavior, and future research could further the discussion of mixed emotions using more appropriate measures of emotions or questionnaires with emotions.

Mixed emotions exist in different cultural backgrounds, and mixed emotions with the influence of local culture and related policies may also have an impact on consumer behavior change. Therefore, this may be explored in future research by replicating the study in multiple cultural backgrounds. It is also important to consider that different cultural backgrounds have significant effects on different psychological aspects, but whether such effects can be replicated across cultures requires further consideration in future research. It would be interesting for future research to discuss whether there is an effect of consumer mixed emotions on consumer improvisation behavior in different cultural backgrounds, given the age and educational background factors. It is noteworthy that other studies could be conducted to explore which factors induce awe and anxiety that influence improvisational behavior, which provides an idea for future research directions. It is also possible to test whether the predictions of this paper are replicated in situations where feelings of awe and anxiety are episodic, for example when such feelings of awe and anxiety are not triggered by specific materials, and how consumer improvisation would be different in situations where they occur unexpectedly in everyday life. One question that could be explored in future research is whether there are sequential entry effects for different emotions. Differences in the order of entry may mean that some emotions occur more quickly and for different durations than others, and these differences may be influential in observing unique behavioral trends in mixed emotional states, which are aspects that could be explored further in the future.

## Figures and Tables

**Figure 1 ijerph-19-17076-f001:**
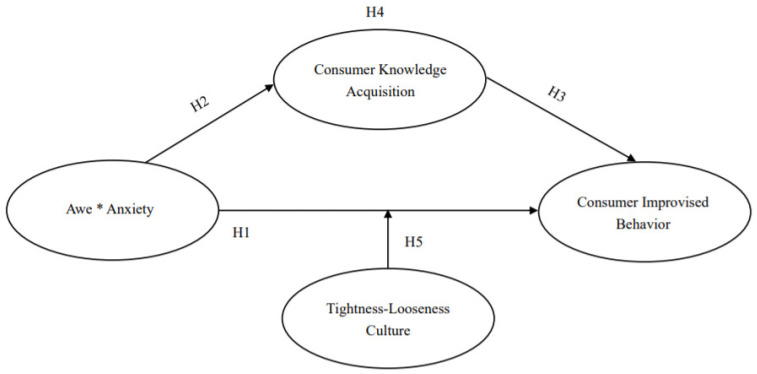
Model Framework Diagram. Note: Awe * Anxiety represents the interaction of the two variables of awe and anxiety.

**Figure 2 ijerph-19-17076-f002:**
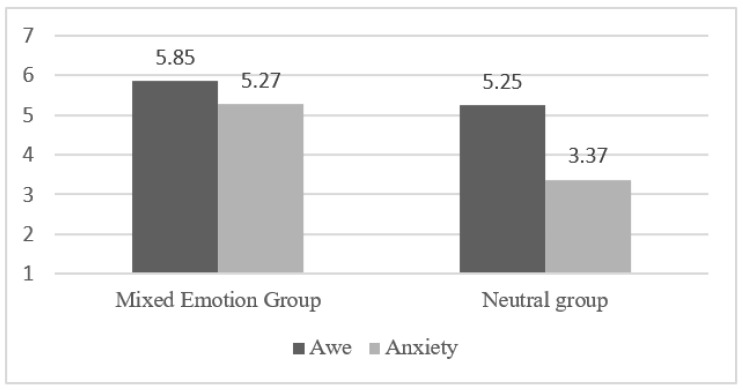
Graph of mean values of awe and anxiety for both groups.

**Figure 3 ijerph-19-17076-f003:**
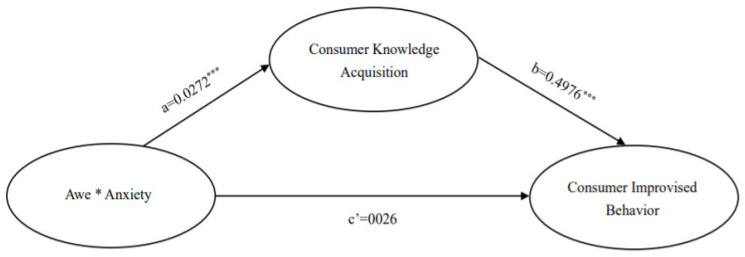
Mediating effect analysis. Note: Awe * Anxiety represents the interaction of the two variables of awe and anxiety; *** indicates *p* < 0.001.

**Figure 4 ijerph-19-17076-f004:**
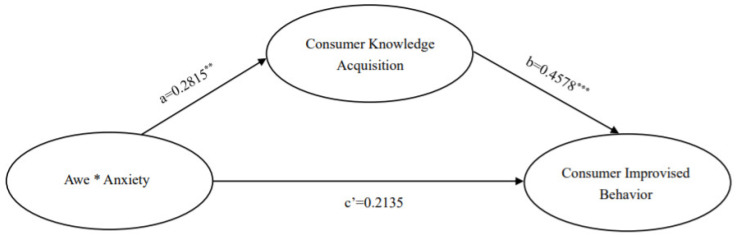
Results of the analysis of mediating effects of consumer knowledge ac-quisition (Looseness culture). Note: Awe * Anxiety represents the interaction of the two variables of awe and anxiety; ** indicates *p* < 0.01; *** indicates *p* < 0.001.

**Figure 5 ijerph-19-17076-f005:**
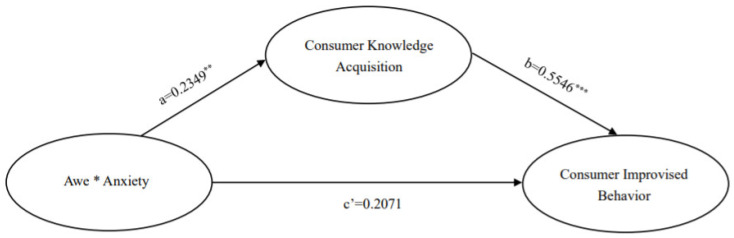
Results of the analysis of mediating effects of consumer knowledge ac-quisition (Looseness culture). Note: Awe * Anxiety represents the interaction of the two variables of awe and anxiety; ** indicates *p* < 0.01; *** indicates *p* < 0.001.

**Figure 6 ijerph-19-17076-f006:**
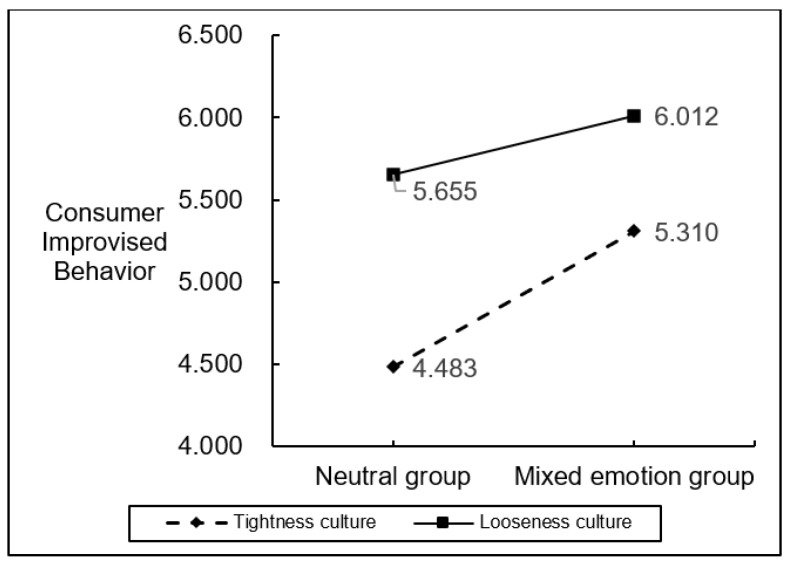
The moderating role of tightness-looseness culture in the relationship between mixed emotions of awe and anxiety and consumer improvisational behavior.

**Table 1 ijerph-19-17076-t001:** Scale of measured variables.

Variables	Title Item	Source of the Scale
Awe	1. In my daily life, I often feel in awe	(Piff, 2015)
	2. In life, I have many opportunities to see the beautiful nature	
	3. In life, I seek challenging experiences where I can make sense of the world	
Anxiety	1. I felt more nervous and anxious than usual	(Zung, 1971)
	2. I feel distracted or feel panic	
	3. I feel like I’m scared for no reason	
Consumer Knowledge Acquisition	1. I am willing to learn and get new and important information from outside	(Parra-Requena et al., 2015)
	2. I am willing to learn and acquire key competencies from outside	
	3. I am willing to learn externally and improve existing capabilities	
Consumer Improvised behavior	1. I am able to deal with unforeseen events immediately	(Vera and Crossan, 2005)
	2. I can improvise when I act	
	3. I can respond immediately to unexpected problems	
	4. I will try to solve problems in new ways and discover new solutions	
	5. I am good at spotting opportunities	
	6. I am willing to take risks when trying new ideas	
	7. Show originality in my approach when solving problems	
Tightness- Looseness Culture	1. In country Y, there are many social norms that people should follow	(Gelfand et al., 2011)
	2. In country Y, if someone misbehaves, others will strongly disagree	
	3. People in country Y almost always follow social norms	
	4. In country Y, there are very clear expectations of how people should behave in most situations	
	5. In country Y, for the most part people agree on what behavior is appropriate and what behavior is not	
	6. People in country Y do not have a lot of freedom in most cases to decide how they want to behave	

**Table 2 ijerph-19-17076-t002:** Analysis of hypothetical effects.

Specific Path	β	SE	LLCI	ULCI
Effect of mixed emotions on knowledge acquisition	0.0272 ***	0.0046	0.0181	0.0363
Effect of knowledge acquisition on improvisational behavior	0.4976 ***	0.0494	0.4005	0.5947
Direct effect of mixed emotions on improvisational behavior	0.0026	0.0044	−0.0060	0.0112
Indirect effect of mixed emotions on improvisational behavior	0.0135 ***	0.0028	0.0086	0.0196
Total effect of mixed emotions on improvisational behavior	0.0161 ***	0.0047	0.0068	0.0254

Note: *** indicates *p* < 0.001.

**Table 3 ijerph-19-17076-t003:** Model path effects table.

Specific Path	β	SE	t	LLCI	ULCI
Effect of mixed emotions on improvisational behavior	0.0037	0.0063	0.5861	−0.0087	0.0160
Effect of mixed emotions on knowledge acquisition	0.0198 **	0.0034	5.7709	0.0131	0.0266
Effect of knowledge acquisition on improvisational behavior	0.5895 ***	0.0453	13.0189	0.5005	0.6786
Effect of mixed emotions * tightness-looseness culture on improvisational behavior	0.0193 ***	0.0086	2.2580	0.0025	0.0362

Note: mixed emotions * tightness-looseness culture means interaction of two variables; ** indicates *p* < 0.01; *** indicates *p* < 0.001.

## Data Availability

The data associated with this study are available on reasonable request from the corresponding author.

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
