# Peer review of "The Impact of Mixed Emotions on Consumer Improvisation Behavior in the Environment of COVID-19: The Moderating Effect of Tightness-Looseness Culture"

_ijerph, 2022, doi:10.3390/ijerph192417076_

Round 1

Reviewer 1 Report

The introduction needs to be re-shape again by enhancing `the background of the study, research objective, and motivation. In the Research method, the authors did not explain how to choose the sample/respondents. Are there any criteria? Perhaps ages or background of education. in Research findings, did the author analyze based on education or age? I think this is an interesting part too since this study was only conducted in one country.  Please stated the authors' contributions to this study.

Author Response

Thank you for your careful review of my manuscript, and for your excellent suggestions on how I can improve the paper. I show below how and where I have responded to your comments. (I have paraphrased your comments in bolded italics.)

R1 #1. The introduction needs to be re-shape again by enhancing `the background of the study, research objective, and motivation.

Response: Thank you for your suggestions.

In the introduction section, we enhance the background, research objectives and motivation of this paper by cutting through the research perspective and then expressing theoretical advances to existing research, while pointing out the insights that would help managers. (See Introduction)

R1 #2. In the Research method, the authors did not explain how to choose the sample/respondents. Are there any criteria? Perhaps ages or background of education.

Response: Thank you for your reviews

In the selection of the sample for the study methodology, we had no special criteria for the sample as we took into account the inclusiveness of the looseness-tightness culture. There were attention check questions during the completion of the questionnaire and data from participants who did not pass the attention check questions were excluded from the study.

R1 #3. in Research findings, did the author analyze based on education or age? I think this is an interesting part too since this study was only conducted in one country. Please stated the authors' contributions to this study.

Response: Thank you for your comments.

In the analysis of the results, we analysed the data using age as a control variable to describe the mean age of the sample. Also education level and age are indeed interesting factors to study, and we have added consideration of age and education in the limitations and future research directions, where future research could further explore the impact of age and education on the consumption behavior of people from different cultural backgrounds. (See Research limitations and future research directions)

In the introduction, we revise and add to the theoretical contributions and management implications. Based on the archetypal theory of awe and the theory of terror management, this paper takes the mixed emotions of awe and anxiety as a research perspective, firstly extending the previous research on improvisation at the organizational level to the individual level and theoretically enriching the existing research literature on individual improvisation. Secondly, it draws on some scholars' extension of tightness-looseness culture to consumer behavior research to shed new light on the mechanisms by which mixed emotions affect consumer improvisation. Finally, it is noted that focusing on the role of mixed emotions among consumers helps managers' practical insights and provides support for specific corporate marketing activities. 

Reviewer 2 Report

This paper is well-written. Moreover, the issues are strong enough and I believe this paper needs to be published as soon as possible. I only have few comments as follow: 

Abstract – add the number of participants and the sampling method (convenience or purposive sampling).

The last paragraph of the introduction part is too long. An extra effort is required to enhance the readability of the manuscript.

This study may cite and discuss these studies in the previous studies since it is highly connected to this study:

·       Prasetyo, Y. T.; Castillo, A. M.; Salonga, L. J.; Sia, J. A.; Seneta, J. A. Factors affecting perceived effectiveness of COVID-19 prevention measures among Filipinos during enhanced community quarantine in Luzon, Philippines: Integrating Protection Motivation Theory and extended theory of planned behavior. International Journal of Infectious Diseases 2020, 99, 312–323.

·       Yuduang, N.; Ong, A. K.; Vista, N. B.; Prasetyo, Y. T.; Nadlifatin, R.; Persada, S. F.; Gumasing, M. J.; German, J. D.; Robas, K. P.; Chuenyindee, T.; Buaphiban, T. Utilizing structural equation modeling–artificial neural network hybrid approach in determining factors affecting perceived usability of mobile mental health application in the Philippines. International Journal of Environmental Research and Public Health 2022, 19, 6732. 

Author Response

Thank you for your careful review of my manuscript, and for your excellent suggestions on how I can improve the paper. I show below how and where I have responded to your comments. (I have paraphrased your comments in bolded italics.)

R2 #1. Abstract – add the number of participants and the sampling method (convenience or purposive sampling).

Response: Thank you for your suggestions.

We have added the number of subjects and sampling method to the abstract to give the reader a clearer understanding of the article.

R2 #2. The last paragraph of the introduction part is too long. An extra effort is required to enhance the readability of the manuscript.

Response: Thank you for your comments.

We have simplified and trimmed the last paragraph of the introduction by indicating the research perspective in the last paragraph of the introduction and then expressing the theoretical advancement of existing research while pointing out management implications to improve the readability of the manuscript.

Reviewer 3 Report

The paper appraises very interesting subject of the impact of mixed emotions on consumer improvisation behavior in the pandemic time.

In my opinion, the article is well written from scientific point of view. The main strengths of the article are the following:

·    clear article structure,

·  correctly formulated hypotheses, which were then verified with adequate statistical methods,

·     comprehensively described research methodology,

·      scrupulous analysis of research results,

·   shortcomings of the research and planned research continuation in the wider area.

However, it is advisable to supplement the article with information about the time and period of the research. The author should also reread the article and correct any linguistic shortcomings and typos.

Author Response

Thank you for your careful review of my manuscript, and for your excellent suggestions on how I can improve the paper. I show below how and where I have responded to your comments. (I have paraphrased your comments in bolded italics.)

R3 #1. However, it is advisable to supplement the article with information about the time and period of the research.

Response: Thank you for your suggestions.

We have added the relevant study times to the study content section to give a more complete picture of the timeline of the research experiments. (See Research Content and Methodology)

R3 #2. The author should also reread the article and correct any linguistic shortcomings and typos.

Response: Thank you for your opinions.

We have reread the article to correct grammatical issues and typos, and used a professional proofreader to proofread it. In this response letter, I asked the editor to help me recommend a professional copy editor if you and/or the editor still feel further copy-editing is needed.

Reviewer 4 Report

To support the reviewed manuscript, I am sending specific comments:

1. In paper explores the Impact of Mixed Emotions on Consumer Improvisation Behavior in The Environment of COVID-19.

The theoretical model proposed by the authors was based on their own research.

The main research questions are:

How consumers' mixed emotions arising from facing unexpected events in an uncertain environment affect their own improvisational behavior after experiencing COVID-19?

2. The topic described in the manuscript is very important and requires constant research. Mixed emotions exist in different cultural backgrounds, and mixed emotions with the influence of local culture and related policies may also have an impact on consumer behavior change. Therefore, may be explored in future research by replicating the study in multiple cultural backgrounds.

3. Compared to other publications, the research provides a new theoretical tool for studying mixed emotions. In addition, they also relate to practical problems, so it is of great theoretical and practical importance.

4. The research methodology is very clear, understandable and very well presented. The reviewer has no comments. The hypotheses are correct and compatible. The test results are clearly presented.

5. The conclusions are consistent with the evidence and arguments and relate to the intended purpose of the study.

6. The references used in the manuscript are appropriate and current. 62 references were used.

Considering the above, I recommend the article to be published in the IJERPH.

Author Response

Thank you for your careful review of my manuscript, and for your positive assessment that this paper is interesting and valuable.

It has been three years since the outbreak of COVID-19, There is less relevant literature on individual responses to emergencies and it is not clear what happens at the individual level. As you have commented, this paper explores the impact of mixed emotions on consumer improvisation behavior in the environment of COVID-19. The main research questions are: How do consumers' mixed emotions arising from facing unexpected events in an uncertain environment affect their improvisational behavior after experiencing COVID-19?

The behavioral response of consumers with mixed emotions in the face of unexpected situations is the content of this paper. In this paper, the study was conducted in China only, however, mixed emotions exist in different cultural backgrounds, and mixed emotions with the influence of local culture and related policies may also have an impact on consumer behavior change. Therefore, may be explored in future research by replicating the study in multiple cultural backgrounds.